# The Role of Stimuli-Driven and Goal-Driven Attention in Shopping Decision-Making Behaviors—An EEG and VR Study

**DOI:** 10.3390/brainsci13060928

**Published:** 2023-06-08

**Authors:** Farzad Saffari, Sahar Zarei, Shobhit Kakaria, Enrique Bigné, Luis E. Bruni, Thomas Z. Ramsøy

**Affiliations:** 1Neurons Inc., 2630 Hoje-Taastrup, Denmark; s.zarei.s@gmail.com (S.Z.); thomas@neuronsinc.com (T.Z.R.); 2Augmented Cognition Lab, Aalborg University, 2450 Copenhagen, Denmark; leb@create.aau.dk; 3Department of Psychology, University of Copenhagen, 1172 Copenhagen, Denmark; 4Faculty of Economics, University of Valencia, 46010 Valencia, Spain; shobhit.kakaria@uv.es (S.K.); enrique.bigne@uv.es (E.B.)

**Keywords:** EEG, virtual reality, attention, coherency, spectral entropy

## Abstract

The human attention system, similar to other networks in the brain, is of a complex nature. At any moment, our attention can shift between external and internal stimuli. In this study, we aimed to assess three EEG-based measures of attention (Power Spectral Density, Connectivity, and Spectral Entropy) in decision-making situations involving goal-directed and stimulus-driven attention using a Virtual Reality supermarket. We collected the EEG data of 29 participants in 2 shopping phases, planned and unplanned purchases. The three mentioned features were extracted and a statistical analysis was conducted. We evaluated the discriminatory power of these features using an SVM classifier. The results showed a significant (*p*-value < 0.001) increase in theta power over frontal, central, and temporal lobes for the planned purchase phase. There was also a significant decrease in alpha power over frontal and parietal lobes in the unplanned purchase phase. A significant increase in the frontoparietal connectivity during the planned purchase was observed. Additionally, an increase in spectral entropy was observed in the frontoparietal region for the unplanned purchase phase. The classification results showed that spectral entropy has the highest discriminatory power. This study can provide further insights into the attentional behaviors of consumers and how their type of attentional control can affect their decision-making processes.

## 1. Introduction

The human attention system has been traditionally divided into two modes. At any given moment, our attention has the ability to shift between external stimuli in our environment and internal cognitive processes, such as memories [1]. Our attention systems have the capacity to be directed toward internal thoughts, such as our memories or plans [2], or can be captured by an external stimulus. Based on this division in cognition, attention is typically distinguished as belonging to one of two types: bottom-up (i.e., externally directed/stimuli-driven/exogenous) and top-down (i.e., internally directed/schema-driven/goal-driven/endogenous) attention [1,3,4].

People rely either on external or internal information when making decisions [2,5]. The decision maker’s goals, intentions, and previous knowledge drive top-down attention. By contrast, decision-making based on bottom-up attention is determined by the lower-level perceptual properties of a stimulus. Additionally, recent studies demonstrate that the bottom-up process is more activated during human “free” choices, where decisions are made without any external forces [6,7,8].

For decades, electrophysiological research has been assessing the underlying neural mechanisms of attention processes [9,10,11,12,13]. Two major brain oscillatory rhythms have been studied in electroencephalogram (EEG) for their functional role in attentional processes: theta (roughly between 4 and 8 Hz) and alpha (roughly between 8 and 12 Hz) [13,14]. Power modulations of these oscillations have been found to be strongly correlated with attentional state alteration in specific regions, as well as interregional synchronization within these frequency bands (measuring functional connectivity) [15,16,17,18]. We briefly review recent studies here that concern the modulation of EEG (particularly in the theta and alpha bands) in top-down and bottom-up attention.

### 1.1. Theta Activity

Theta activity has been observed mainly in the frontal midline region (around Fz) in goal-driven tasks which involve internal attention, such as internal planning [19] and working memory tasks [20,21,22]. In order to retrieve information from multiple items/cues, the mentioned tasks require integrating, updating, organizing, and holding information. These studies suggest that theta activity is associated with internally directed attention toward information/intention stored in the memory and that it helps with information retrieval [13,16]. The evidence from theta oscillations mediating the orchestration between the temporal lobe and medial prefrontal cortex (MPFC) also supports its role in memory integration [21]. Interestingly, an increase in theta activity in the MPFC is also observed in prospective memory tasks [23], which is remembering to execute the planned task and ignoring the irrelevant stimuli [24,25]. This requires orienting attention toward internal intentions in the memory and successful retrieval of that information. In addition, an increase in the theta synchronization between frontal, temporal, and posterior regions of the brain has been observed in those tasks [22].

### 1.2. Alpha Activity

Alpha-band oscillations regulate attention processes both inside and outside of the focus of attention. The most prominent role of alpha-band oscillations in attention is the inhibition of task-irrelevant information that can interfere with task goals [15,26] and enhancing the processing of relevant information. This function is important for the selection and suppression processes required in attention and therefore modulates knowledge access and orientation [27,28]. Moreover, an increase in alpha activity has been found during retention intervals in visual working memory tasks [29]. The increase in alpha power in memory tasks is associated with task difficulty and memory load, which itself supports the inhibitory role of alpha for the maintenance and protection of task-relevant information and filtering external interrupting information [15,27,29]. As a filter mechanism, through a progressive increase in power, alpha oscillations inhibit the distractors. The higher the attention to task-relevant stimuli, the higher the suppression of distracting information [27]. Therefore, the alpha increase has been associated with internal attention, while external attention has been linked mainly to alpha decrease [26,30,31].

More recently, there has been a rise in more advanced methods for studying and understanding the brain mechanisms of attentional processes. Here, we focus on two analysis methods that also represent underlying brain mechanisms that may be relevant for understanding attentional processes in the brain: synchronous brain activity, and entropy-based measures of brain activity.

### 1.3. Synchronized Intra-Regional Activity

The dorsal frontoparietal attention network is one of the executive networks that mediate goal-directed behaviors [4,32,33]. A combination of this inhibitory mechanism and an excitatory baseline shift that orients attention sharpens visual spatial focus [34]. Studies that have assessed connectivity between parietal and dorsofrontal attentional regions have shown that voluntary vs. involuntary attention have different attentional networks (the dorsal attention networks) depending on the type of attention. The role of bottom-up and top-down processes in action comprehension has been investigated in previous studies [35]. They suggest that there exists a distinct interactivity among brain regions indicating the degree of bottom-up vs. top-down attention. In addition to that, a large amount of research has suggested that synchronized activity in the aforementioned networks in the theta frequency band is modulated by internally directed attention and sustained attention [22,36,37,38,39,40]. Having said that, an increase in low-frequency coupled activity in the frontoparietal network has been considered an indicator of working memory and top-down brain activity [22,36]. However, these well-established concepts and neural behaviors have not been discussed or investigated in the consumer neuroscience context. Particularly, to what extent the underlying neural response of humans’ attentional process could explain consumers’ shopping behavior is still missing.

### 1.4. Spectral Entropy

Entropy (e.g., Shannon entropy) is a concept in stochastic signals that quantifies the irregularity of random variables by measuring proportion distribution [41]. Since its first debut in EEG [42], Shannon or spectral entropy (SpE) has been used as an irregularity index of EEG signals where higher values correspond to more irregular or unpredictable signals (more “flat” distribution) and lower values correspond to more predictable signals (power spectrum is bounded in specific frequency band) [43]. Recently, spectral entropy has been utilized as an attention index [43] to develop a more accurate attention-based diagnostic tool. Moreover, modulation of SpE in focused attention when exposed to auditory stimuli has been investigated, and a decrease in SpE has been observed in active attention compared with passive attention in response to audio stimuli [44,45]. To our knowledge, only one study of attention to visual objects has been published [46], where the researchers report a greater approximate entropy for externally operative attention compared with internally operative attention. In spite of this discussion around the role of entropy in the human attention system, it has not been tested whether the activation of different attention modalities such as top-down or bottom-up will modulate SpE.

Extended reality technologies such as Virtual Reality (VR) have been found to be useful in marketing, retail, and consumer behavior analyses, such as of attention [47,48,49,50]. VR has a multidimensional framework with real-time graphics. Interactivity, imagination, and immersion are the essential features of VR; therefore, using VR for testing consumer behavior provides the participants with a 3D dynamic purchase experience that is closer to real life [47,51,52].

Previous studies that have explored the relationship between brain oscillations and attentional control have generally not focused on consumer behavior. Furthermore, although numerous studies have investigated the effect of attention level on the SpE of EEG, to the best of our knowledge, the effect of goal-driven and stimulus-driven attention on SpE has not been previously examined. Additionally, no prior study has assessed brain activity related to different modes of attention in (a VR) environment, making the present study particularly ecologically valid. To understand the impact of distractors in the environment or visual field, it is also important to investigate connectivity and power across brain regions when attention is directed toward task-relevant stimuli versus when it is directed toward distractors or external stimuli.

Therefore, the aim of the present study was to assess three types of EEG-based measures of attention and their discriminatory power in decision-making situations involving goal-directed and stimulus-driven attention using a VR supermarket. The experiment consisted of two phases: a planned purchase phase (or listed condition, in which participants shopped from a list) and an unplanned purchase phase (in which participants were free to buy what they wanted). We hypothesized that the unplanned purchase phase would elicit more bottom-up attention, while the planned purchase phase would elicit more top-down attention. Accordingly, we expected to see higher theta activity over frontal, central, and temporal lobes during the planned purchase phase and lower alpha activity in frontal and parietal lobes during the unplanned purchase phase. Furthermore, during the planned purchase phase, we predicted a higher level of synchronized activity within the frontoparietal network Additionally, we hypothesized that there would be a more distinct alternation of SpE in the frontal, parietal, and occipital lobes, reflecting shifts between goal-directed and stimulus-driven attention. At a data analysis level, this means that we expected to observe an increase in SpE in the frontal, parietal, and occipital lobes, from planned to unplanned purchase phases. Furthermore, we evaluated the discriminatory power of the three mentioned features using an SVM classifier.

## 2. Materials and Methods

### 2.1. Participants

A total number of 29 (14 women and 15 men) volunteers (age range: 23 to 44; mean = 31.8; SD = 6.6) without any prior psychiatric or neurological conditions were recruited in the experiment via the Neurons Inc. online recruitment system. Among these participants that we considered as the data for the study, eight of them had previous experience with VR and using controllers and two of them had participated in a study with a virtual supermarket previously. All participants were informed about the experiment and had read and signed the consent form prior to the experiment. The experiment was approved by the ethical committee of the University of Aalborg. All data were analyzed and reported anonymously.

### 2.2. Experimental Procedure

The experiment was performed in a virtual reality environment designed in Unreal Engine V4.1 and run using HTC Vive 5 in a machine with 16 GB of RAM (Intel(R) Core (TM) i7-10875H CPU 2.30 GHz) and NVIDIA GeForce RTX 2070 GPU on Microsoft Windows 10 operating system. A supermarket was designed in VR (similar to Danish supermarkets), and we asked participants to perform a shopping task. We allocated 250 Danish Kroner, DKK (~USD 35), to each participant. We first asked them to buy six items from a predefined list (the planned purchase phase), and then they could buy whatever they wanted with the remaining budget (the unplanned purchase phase). The cost of the items on the list added up to DKK 120, which left half of the budget for unplanned purchases. Overall, 172 items were chosen as unplanned purchases, which is almost equal to 174 planned purchases.

First, we collected 30 s of resting-stage EEG data (with eyes closed and a black screen) while the participants had the VR headset on. Afterward, we instructed participants on how to use the controller to navigate through the supermarket, find the list of products, and choose a product. In the physical space, participants could have limited movement (one or two steps for getting closer to a product) but in general, they were instructed to teleport with the controller to navigate through the supermarket, which they could do by pointing to a location and reaching there with a short delay (200 ms), to ensure they would experience a “flow” in the supermarket. The list of the required products was provided for them in VR and participants could look at the list whenever they needed using a button in the controller. The products had price labels and the prices were according to the actual price ranges for products in Danish supermarkets.

When participants were completely familiar with the tasks, we started to record the data, and they needed to first purchase six items from the list (i.e., broccoli, milk, cheese, soda, cereal, and chocolate). This phase was considered as the “Planned Purchase Phase”. We expected “goal-driven” or “top-down” attention to be activated in this phase. When the participants had purchased all the items on the list, we asked them to buy whatever they wanted with the remaining budget, or they could leave the environment immediately. The remaining time that the participants spent buying the items they wanted using the leftover money was considered the “unplanned purchase phase”. We expected “stimuli-driven” or “bottom-up” attention to be activated in this phase. After participants were done with the shopping task, they had to go to the cashier, which was taking them out of the environment.

### 2.3. EEG Recording and Processing

EEG data were recorded via Brain Product EEG device with 32 electrodes (Fp1, Fp2, F8, F4, Fz, F3, F7, FT9, FT10, FC5, FC1, FC2, FC6, T7, T8, C3, Cz, C4, CP5, CP1, CP2, CP6, TP9, TP10, P7, P3, Pz, P4, P8, O1, Oz, and O2). The ground and reference electrodes were located at AFz and FCz, respectively, for online processing. The data were transmitted wirelessly from the amplifier to a PC using a USB module. The EEG data were digitalized with a 500 Hz sampling frequency and then exported to the MNE Python library for pre-processing and further analysis. For each participant, 30 s of the rest data were recorded, but the time spent for each condition varied among participants. The following pre-processing steps were applied for the EEG data of each participant (the whole analysis for the data from pre-processing to statistical analysis was carried out via different Python libraries). First, the data were filtered using an FIR bandpass filter with a hamming window for the 0.1 and 100 Hz frequency bands, and then a 50 Hz notch filter was applied to remove the power line artifact.

Independent Component Analysis (ICA) was used to manually remove “bad” components with a visual inspection. The components that contain the eye-blink and eye-movement patterns were eliminated based on visual inspection. Furthermore, components with an increasing pattern of power distributions with regard to the frequency were removed during ICA. On average, 9.5 components out of 32 were removed during ICA for each subject. Thereafter, we changed the EEG reference to average all electrodes for the rest of the analysis. Then, we segmented data into 5 s epochs for consistency of the analysis and computational convenience.

### 2.4. Power Spectral Density Computation

Power spectrum density (PSD) was computed via the Welch method with a window size of 256 samples, equal to 512 milliseconds (ms) for each phase (i.e., rest, planned purchase, and unplanned purchase). Theta (4–8 Hz) and alpha (8–13 Hz) frequency bands were considered for PSD analysis. For each channel, first, we averaged power values over frequency bins, and then we took the average over epochs to represent the PSDs of that condition for the corresponding channel. To compute PSD over regions, we averaged the PSD of channels included in those regions. The underlying regions of each EEG channel are shown in Table 1.

### 2.5. Functional Connectivity

Spectral coherency is a synchronization metric that has been widely used to measure functional connectivity in neuroscientific studies [53,54,55,56,57]. As is shown in Equation (1), coherency shows variance explainability of signal *X* with regard to signal *Y* by measuring cross-spectral density of *X* and *Y* ($S_{XY}$), normalized by power estimates of *X* and *Y* ($S_{X},S_{Y}$).

Equation (1):(1)$${Coh}_{XY}\left(\omega \right)=\frac{\left|S_{XY}\left(\omega \right)\right|}{\sqrt{S_{XX}\left(\omega \right)S_{YY}\left(\omega \right)}}$$
where coherence coefficient is a value between 0 and 1; 0 indicates no synchronization, and 1 indicates perfectly synchronized signals.

Due to spurious connectivity caused by volume conduction [54,58], we conducted current source density [59,60] to reduce the volume conduction effect. Afterward, we used the Fourier method to transform the data from a time domain to a frequency domain with 256 points (equal to 512 ms) in the theta frequency band (4–8 Hz). Thereafter, by computing coherence between two given signals, we averaged coherency measurement over frequency bins to represent the connectivity values for each of the two channels. Lastly, by averaging pairwise connectivity values in regions of interest (especially the frontoparietal networks), the final connectivity values were computed and ready for statistical analysis.

### 2.6. Spectral Entropy

In information theory, entropy and spectral entropy are analytical techniques to quantify irregularity and (un)predictability in a stochastic signal such as EEG [41]. As noted in Equation (2), spectral entropy (SpE) uses power spectrum density to quantify EEG irregularity between 0 and 1, where ‘0′ means no irregularity (totally predictable) and ‘1′ means a completely random sequence.

Equation (2):(2)$$SE\left[f_{1}{,f}_{2}\right]=\frac{1}{log\left(N\left[f_{1}{,f}_{2}\right]\right)}\sum_{f_{i}=f_{1}}^{f_{2}}P_{n}(f_{i})\cdot log\left(\frac{1}{P_{n}(f_{i})}\right)$$
where $P_{n}(f_{i})$ represents normalized power spectrum at the frequency of $f_{i}$, which, by dividing the power spectrum at each frequency by total power spectrum, yields the normalized power spectrum. $f_{1}$ and $f_{2}$ are the boundaries of the frequency range, and $N\left[f_{1}{,f}_{2}\right]$ is the number of frequency bins within that range.

We considered a frequency range of 0.5 to 32 Hz and used FFT (Welch’s method) to compute the PSD with a window of 256 points, which is equal to a 512 ms time window. In addition, since attention mainly activates frontal and parieto-occipital networks [4], our focus for entropy analysis was mainly on these areas.

### 2.7. Statistical and Classification Analysis

As noted, the time spent in each condition varied among the participants. Therefore, in order to have a valid comparison, we used a permutation-based statistical method that previously had been used in an ERP study for imbalance trial comparison [61]. First, desired features (PSD, connectivity, and SpE) were calculated from the trial of each condition. Then, those values of one condition were subtracted from the other to calculate the difference between the features. By repeating this procedure for all subjects, the average of the relevant feature was computed to provide the “ground truth” values. Next, for generating data-driven null distribution, we randomly shuffled trail labels between conditions and repeated the same procedure of calculating the ground truth. By repeating this procedure 1000 times, we conducted a null distribution to compare the ground truth value in the two conditions. Ultimately, if the ground truth was far enough from the mean of this distribution, we could conclude that our findings are not due to randomness and are statistically significant.

To compare the discriminatory power of PSD, connectivity, and SpE in classifying goal-directed or stimulus-driven attention, we implemented a support vector machine (SVM) classifier [62] and used these features as inputs. In addition, by measuring the Area Under the Curve (AUC) of the Receiver Operating Characteristic (ROC) for each feature, the performance of those features was evaluated in a 5-fold cross-validation setting.

## 3. Results

Since the duration of the experiment was subject-dependent, we provided the time length of the data here to show the amount of EEG data we used for the analysis. In Figure 1, box plots of the time that each participant spent on the two phases are provided. The mean duration of the planned and the unplanned purchase phases across participants are 238.87 ± 85.57, and 228.00 ± 107.20 s, respectively.

To evaluate the subjective experience of the VR supermarket, two survey questionnaires with a 7-point Likert chart were administered to inquire about participant satisfaction with the store and their sense of presence. We found a significant correlation between “Sense of presence” and “Store Satisfaction” r (27) = 0.54, *p*-value < 0.01. The results are reported in Figure 2.

As mentioned, participants were free to choose whatever they wanted under the unplanned purchase condition, and therefore, a range of choices were made that differed from fixed purchases under the planned condition. In Figure 3, the distribution of the unplanned purchases is illustrated.

### 3.1. Power Spectrum Analysis

The distribution of theta and alpha power over the scalp during the rest, planned purchase, and unplanned purchase phases are presented in Figure 4 and Figure 5 (averaged across all subjects). Figure 6 presents the regional statistical comparison for the theta frequency band, focusing on the frontal and parietal regions. During the planned purchase phase, there was an increase in Fz activity compared with the rest and unplanned purchase phases. In the fronto-central regions (Fz, Cz, Fc1, Fc2), theta power during the planned phase was higher than during the rest and unplanned purchase phases. In the temporal lobe (T7, T8, TP9, TP10, FT9, FT10), an increase in theta activity was observed under the planned condition compared with the rest and unplanned conditions. This increase was also seen under the unplanned condition compared with the rest. In addition, a similar pattern of variation was observed in T7 and T8.

A statistical comparison of the three conditions for the alpha frequency band in the frontal and parietal lobes is presented in Figure 7. A decrease in alpha power over the parietal lobe occurred for the unplanned phase and was not observed in the rest and planned phases. In the frontal lobe, a stronger decrease in alpha activity was found for the unplanned condition, in comparison with the rest and planned purchase phases. We observed an increase in alpha activity in the planned phase compared with the rest, but it was not statistically significant.

### 3.2. Functional Connectivity Analysis

Table 2 and Figure 8 show the results of a connectivity analysis for the frontal–parietal networks after applying the current source density. In Table 2, the coherency values in the theta frequency band for both the planned and unplanned conditions are presented. For all pairs (frontal–frontal, fontal–parietal, and parietal–parietal), the coherency values for the planned phase were significantly higher (*p*-value < 0.0001, *p*-value < 0.000, *p*-value < 0.001, respectively) than for the unplanned phase.

In Figure 8, the circle of coherency values between and within the frontal and parietal lobes is illustrated. For both phases, coherency values were averaged over all epochs, and for visualization in Figure 8, the mean of coherency values was averaged over all participants. A stronger synchronization, both locally and regionally, was found in the planned purchase phase compared with the unplanned purchase phase.

### 3.3. Spectral Entropy Analysis

The spectral entropy values for 32 channels are presented in Table 3. For each channel, the spectral entropy for the planned purchase phase (Entropy 1), the unplanned purchase phase (Entropy 2), the difference (Entropy 2–Entropy 1), and the *p*-value (with a significance level being 0.05/32 = 0.001) are reported. For Fp1 and F7, sensors that are placed in the frontal region, spectral entropy in the unplanned purchase phase was significantly higher (*p*-value < 0.0001) than in the planned purchase phase. In the parietal and occipital lobes, P8 and Oz channels showed a higher spectral entropy for the planned purchase phase than the unplanned purchase phase, and they were statistically significant (*p*-value < 0.0001). In Figure 9, entropy values and topography plots are provided for the planned purchase phase, the unplanned purchase phase, and their differences.

### 3.4. Classification Results

The results of this subject-independent model are shown in Figure 10 and Figure 11. The highest accuracy was achieved when all three features were used as inputs, and other metrics were also relatively high, indicating a balanced performance of the model in predicting between the two classes. Among the individual features, SpE resulted in the highest accuracy, with comparably high precision, recall, and F1 score, indicating that the model was not biased toward either class. When using PSD as the sole input, the classifier exhibited an imbalanced performance, with the highest sensitivity (91%) but low accuracy and precision, indicating a bias toward the goal-directed attention class. The model performed worst in terms of accuracy when using connectivity as the sole input, but the other metrics showed a balanced performance between the two classes.

To compare these features more thoroughly, we conducted a five-fold cross-validation analysis and calculated the AUC of the ROC for each feature and for all features combined as inputs to the model. As shown in Figure 11, the ROC of the model using all features as input had the highest AUC of 0.94 ± 0.04 (average across five folds). This result was similar to the AUC of the model using SpE alone, which was 0.94 ± 0.05, but with a slightly higher standard deviation. Using connectivity as the sole input resulted in an AUC of 0.80 ± 0.15 on average across five folds, which was slightly higher than the AUC of the model using PSD alone, which was 0.79 ± 0.05.

## 4. Discussions

In this study, we investigated to what extent goal-driven versus stimulus-driven attention is involved in different shopping behaviors, i.e., planned, and unplanned decisions. By labeling different decision types as planned and unplanned, we were able to observe a higher activity in both alpha and theta bands over frontal and parietal lobes in the planned purchase phase compared with the unplanned purchase and rest phases. On the other hand, for unplanned purchases, we observed a decrease in both theta and alpha activity in comparison with planned purchases. Stronger connectivity over the frontoparietal network was found in the planned purchase compared with the unplanned purchase phase. However, in the unplanned condition, SpE was higher in the frontal, parietal, and occipital regions.

### 4.1. Theta Power Changes in Frontal, Central, and Temporal Lobes

Compared with the unplanned and the rest phase, an increase in theta power was observed in the frontocentral regions (Fz, Cz, Fc1, Fc2) during the planned purchase phase. This increase was also observed in the temporal region (T7, T8, TP9, TP10, FT9, FT10). The increase in theta band activity, as compared with the rest, was also observed in the unplanned phase for some temporal regions (T7, T8). These findings are in line with the previous studies that support an increase in theta power during goal-driven attention, mainly over frontal, central, and temporal lobes. More specifically an increase in theta power in the frontal midline regions has been found for tasks that require goal-driven control of attention, such as information retrieval and planning [63,64]. This is consistent with our findings, which show a substantial increase in the theta frequency band in the frontocentral region and signify a goal-directed mode of attention in the planned phase. It is worth noting that an increase in theta power in the mentioned regions has been observed in various cognitive tasks, with a stronger increase for more demanding conditions [65]. Similarly, a stronger theta activity has been reported in the temporal region while performing tasks that require the activation of prospective memory (that is, directing the attention toward intentions stored in the memory to execute the planned intention) [13,66]. An increase in the frontal midline has been also observed in the tasks that require sustained internal attention, such as working memory tasks [22]. One of the explanations for the higher activity of theta during these tasks is the necessity for updating and organizing information, a feature presented in our planned purchase phase.

### 4.2. Alpha Power Changes in Frontal and Parietal Lobes

We observed a weaker alpha activity in the parietal lobe for the unplanned phase in comparison with the planned and the rest phases. Similarly, there was also a lower alpha activity during the unplanned phase in the frontal lobe. However, a stronger alpha activity was observed in both the frontal and the parietal regions for planned purchases compared with unplanned purchases. These findings support the previous studies that suggest that an increase in alpha band activity is observable in goal-driven attention or during the processing of “task-relevant information” [15,26,67,68], whereas lower alpha activity has been observed during “sensory-intake tasks”, which require the processing of external sensory information [26]. An alpha power decrease has been found to be the most prominent feature in tasks that require the direction of attention to external stimuli—in our study, the unplanned purchases—and it has been found over occipitoparietal and frontotemporal regions, as well as over left dorsal frontoparietal regions [1,13]. An increase in alpha-band oscillations, however, has been observed in tasks that require internal attention, or in goal-driven tasks. In fact, alpha activity plays an important role in information processing by inhibiting task-irrelevant information and suppressing the distractors that interfere with the task’s goals.

### 4.3. Increased Synchronized Activity over the Frontoparietal Network in Theta for Goal-Driven Attention

We found stronger connectivity both between frontal and parietal regions and within the regions themselves during planned purchases compared with unplanned purchases. This supports our hypothesis that in the planned purchase phase, the goal-driven attentional network is more activated in comparison with the unplanned purchase phase, which is a stimulus-driven process.

The frontoparietal network has been considered as a control network for internally directed attention [37]. Goal-directed attention modulates an increase in the synchronized activity in the frontoparietal network compared with stimulus-driven attention, according to recent findings in the theta frequency band [22,36,37]. Research also shows that top-down control signals may emerge from the dorsoparietal attention network as well as other higher-order executive regions that mediate behavior directed toward goals [4,69,70].

### 4.4. Spectral Entropy Increase in Frontal and Parietal Sites for Stimulus-Driven Attention

A higher SpE was observed in the frontal and parietal regions during the unplanned condition compared with the planned condition. These results suggest that while participants are in a more predictable situation, such as a planned purchase compared with an unplanned purchase, this regularity pattern has its own neural signature in EEG signals. However, due to a lack of adequate evidence in this field, further research is required to investigate the modulation of SpE in goal-driven and stimulus-driven attention.

As mentioned before, the application of SpE as an objective measurement of attention is relatively a new approach [44,45]. SpE has been utilized as an index for attention level [43,44], in which an increase in SpE is associated with “active attention” compared with “passive attention”, especially in the frontal and the parietal regions. However, to the best of our knowledge, the SpE role has not been investigated for comparison of goal-directed versus stimulus-driven attention.

### 4.5. The Discriminatory Power of SpE Is Almost Equal to the Combination of PSD, Connectivity, and SpE

As the classification results suggested, SpE has the highest discriminatory power among the three features to classify goal-directed and stimulus-driven attention. Considering both sensitivity and specificity, using SpE as an input for the predictive model would result in equal performance while using all of the features combined. In addition, although the PSD feature will lead to the highest sensitivity (recall) since will result in poor specificity, the overall performance (AUC) of the model is worse than when using connectivity to feed the model. It is worth mentioning that the train and test sets of the model derive from different subjects, which results in a subject-independent and more generalized performance.

### 4.6. Limitations

Some limitations are worth mentioning for consideration in future studies. One of the limitations of the present study is the uneven duration of the unplanned and planned purchase phases, which could potentially cause a bias in the results. Even though we tried to reduce the bias with the mentioned statistical method, it should be taken into consideration for future research. Another point is that, since we needed to consider the “budget” that each participant could spend on each purchase, the planned phase was always the first phase of the experiment. Nevertheless, considering that the phases were not very long, we assume that this should not have had a major impact on the findings of the study. Moreover, we had no rest phase before the unplanned purchase phase as it started immediately after the planned phase. Having no pause between the phases, however, contributed to the ecological validity of our study.

## 5. Conclusions

In conclusion, this study analyzed alpha and theta activity during planned and unplanned purchase tasks in a VR environment. The findings revealed distinct neural oscillation patterns associated with different phases of the purchase process. During the planned purchase phase, both alpha and theta powers increased, indicating heightened cognitive engagement. In contrast, the unplanned phase showed a decrease in both theta and alpha activity, suggesting reduced cognitive engagement.

Moreover, our investigation provided insights into the functional connectivity within the frontoparietal network during the planned and unplanned purchase phases. Specifically, we observed greater connectivity within the frontoparietal network during the planned purchase phase compared with the unplanned purchase phase. This finding suggests the involvement of coordinated activity between the frontal and parietal regions, which are crucial for attentional control and decision-making processes. Interestingly, the SpE analysis yielded contrasting results, with higher SpE observed in the frontal and parietal regions during the unplanned purchase phase compared with the planned purchase phase, which shows the capability of SpE in examining the human attention system. This suggests a shift in attentional dynamics, potentially reflecting a transition from goal-directed attention to stimulus-driven attention during unplanned purchasing.

Overall, this study expands our knowledge of the neural mechanisms underlying planned and unplanned purchase tasks, offering insights into attentional behaviors in consumer decision-making. These findings have implications for marketers aiming to influence consumer behavior and guide purchase decisions by understanding the neural mechanisms involved in attentional control and decision-making processes.

Future research directions include exploring larger and more diverse samples to enhance the generalizability of the findings. Additionally, investigating the impact of individual differences, such as personality traits or prior purchasing experiences, could provide a comprehensive understanding of consumer behavior. Incorporating other neurophysiological measures, such as event-related potentials (ERPs) or functional magnetic resonance imaging (fMRI), would offer a more detailed characterization of the brain mechanisms underlying purchase decisions. Furthermore, studying real-world purchasing scenarios and considering contextual factors could provide a more ecologically valid understanding of consumer attention and decision-making processes.

## Figures and Tables

**Figure 1 brainsci-13-00928-f001:**
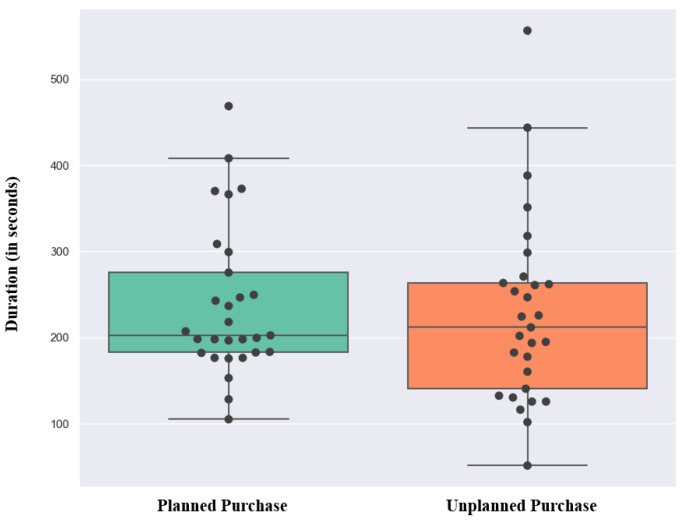
Box plot of the time spent under planned and unplanned conditions for each participant per second. Each dot represents the time spent by each subject and the horizontal line in the boxes represents the median for each condition (202.54 s for planned and 211.86 s for unplanned conditions).

**Figure 2 brainsci-13-00928-f002:**
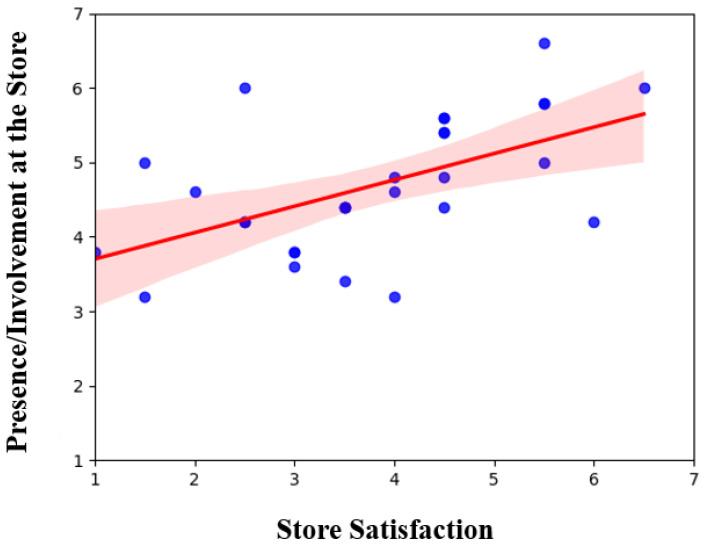
A moderate correlation (0.5) between store satisfaction and sense of presence resulted from the questionnaire.

**Figure 3 brainsci-13-00928-f003:**
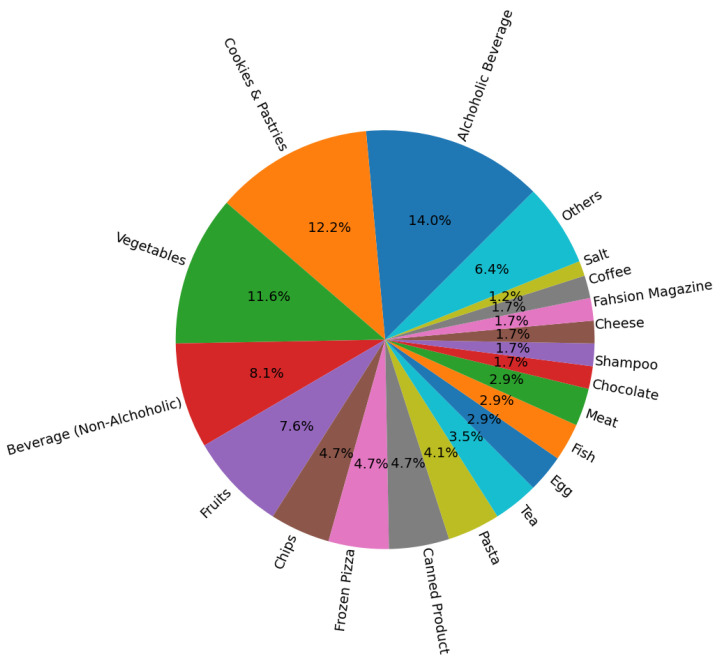
The distribution of product categories that were chosen under unplanned condition.

**Figure 4 brainsci-13-00928-f004:**
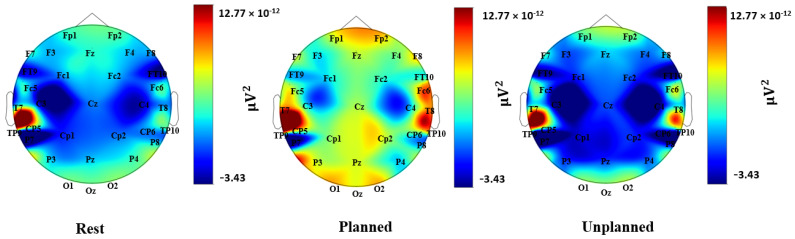
Topographical heatmap of theta activation during the rest, planned, and unplanned phases.

**Figure 5 brainsci-13-00928-f005:**
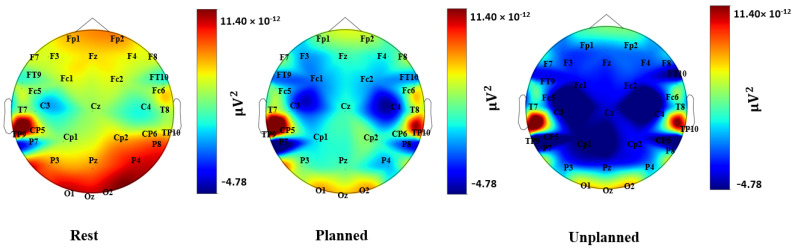
Topographical heatmap of alpha activation during the rest, planned, and unplanned phases.

**Figure 6 brainsci-13-00928-f006:**
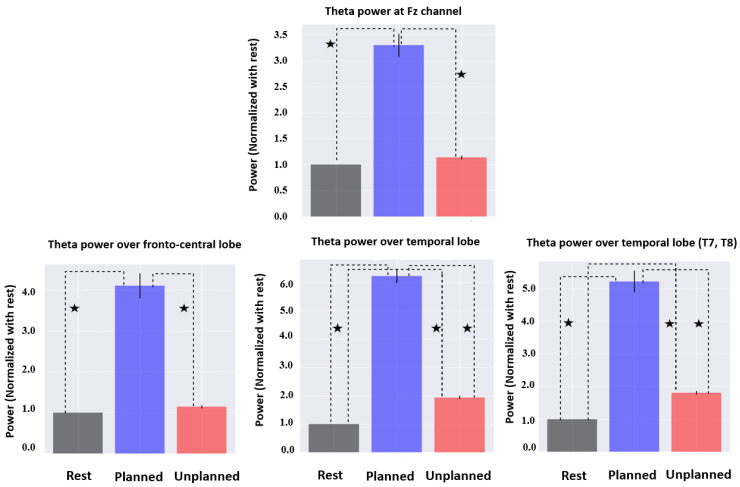
Comparison of theta activity for frontal, central, and temporal regions (normalized by rest). Statistically significant results are shown by ★.

**Figure 7 brainsci-13-00928-f007:**
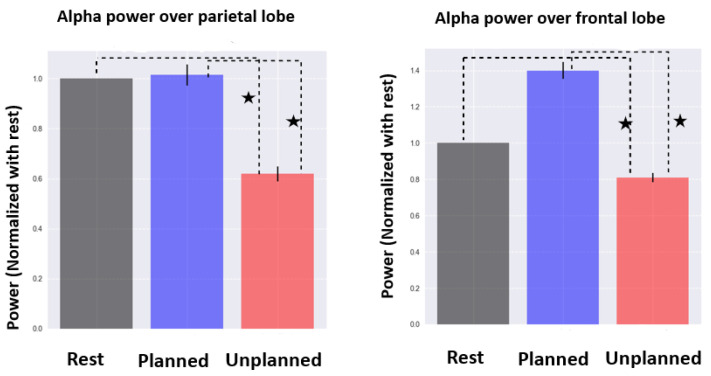
Comparison of alpha activity for frontal and parietal regions (normalized by rest). Statistically significant results are shown by ★.

**Figure 8 brainsci-13-00928-f008:**
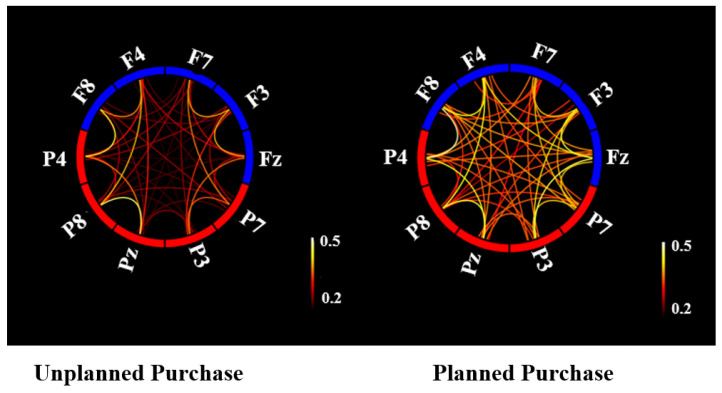
Connectivity circle plots for the planned and unplanned conditions. EEG sensors located in frontal areas are indicated by blue blocks, while sensors in parietal regions are shown by red blocks. The intensity of the color of the links between each block represents the connectivity values between those locations.

**Figure 9 brainsci-13-00928-f009:**
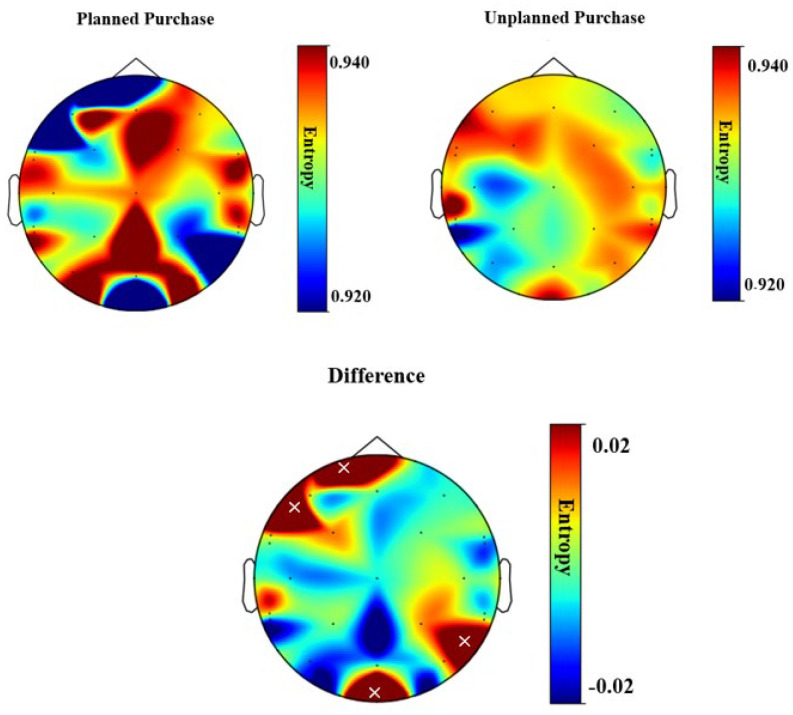
Distribution of SpE values over the scalp for each condition and difference (unplanned and planned). Areas with statistically significant increases in SpE for unplanned purchases are indicated with ×.

**Figure 10 brainsci-13-00928-f010:**
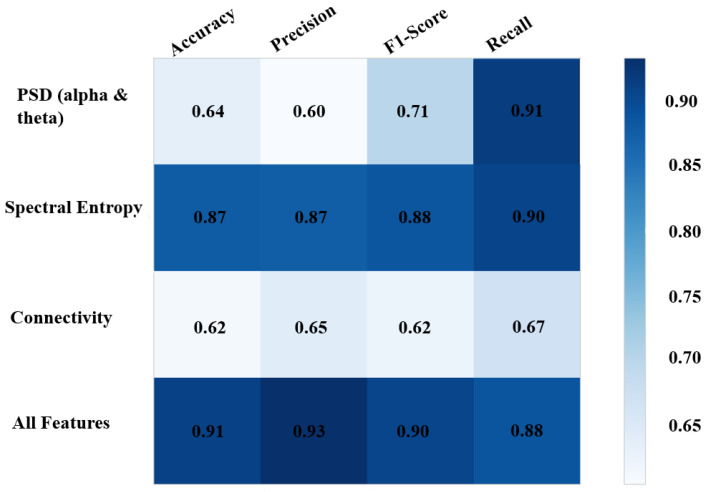
The classification report of the model using different features as input.

**Figure 11 brainsci-13-00928-f011:**
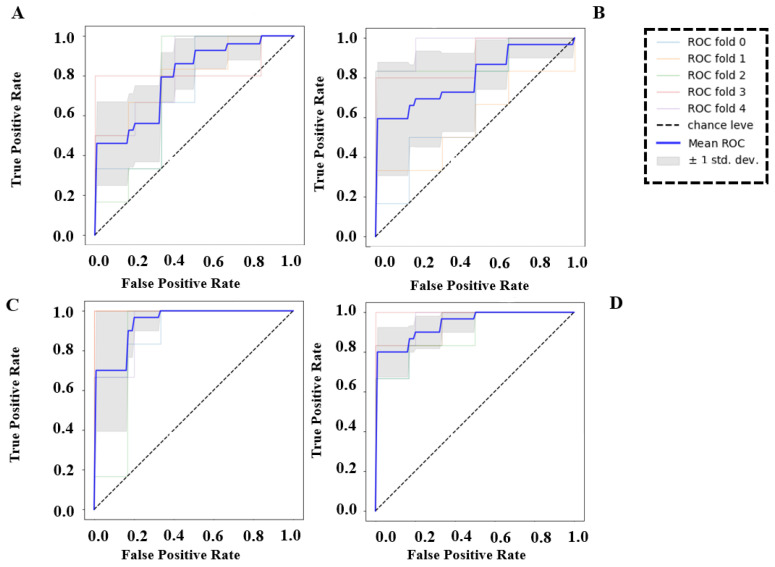
The AUC of ROC in 5-fold cross-validation using (**A**) PSD, (**B**) connectivity, (**C**) SpE, and (**D**) all features as input to the model.

**Table 1 brainsci-13-00928-t001:** EEG channels allocated for each region.

Regions	Channels
Frontal	FP1, FP2, F8, F4, Fz, F3, F7
Parietal	P7, P3, Pz, P4, P8
Temporal	T7, T8, TP9, TP10, FT9, FT10
Occipital	O1, O2, Oz
Central	FC5, FC1, FC2, FC6, C3, Cz, C4, CP5, CP1, CP2, CP6

**Table 2 brainsci-13-00928-t002:** Coherency values (mean ± std) for inter-regional (frontal–frontal, parietal–parietal) and intra-regional (frontal–parietal) for each condition.

Coherency Values			
	Frontal–Frontal	Parietal–Parietal	Frontal–Parietal
Planned Purchase Phase	0.52 ± 0.09	0.51 ± 0.11	0.39 ± 0.10
Unplanned Purchase Phase	0.45 ± 0.08	0.45 ± 0.05	0.31 ± 0.08

**Table 3 brainsci-13-00928-t003:** SpE values of each EEG channel. SpE for the planned and unplanned conditions are indicated with Entropy 1 and Entropy 2, respectively. The Difference column shows the subtraction of Entropy 2 from Entropy 1. Channels with statistically significant different values are indicated with *.

Channels	Entropy 1	Entropy 2	Difference	*p*-Value
*** Fp1**	0.855	0.933	0.078	<0.0001
**Fz**	0.935	0.933	−0.002	0.578
**F3**	0.930	0.934	0.003	0.423
*** F7**	0.843	0.939	0.096	<0.0001
**FT9**	0.936	0.935	0.0	0.567
**FC5**	0.929	0.936	0.006	0.154
**FC1**	0.925	0.937	0.011	0.161
**C3**	0.934	0.926	−0.007	0.655
**T7**	0.935	0.934	−0.001	0.575
**TP9**	0.929	0.934	0.005	0.359
**CP5**	0.936	0.925	−0.01	0.92
**CP1**	0.929	0.929	0.0	0.571
**Pz**	0.934	0.930	−0.003	0.644
**P3**	0.937	0.925	−0.011	0.821
**P7**	0.931	0.929	−0.002	0.522
**O1**	0.934	0.932	−0.002	0.687
*** Oz**	0.815	0.939	0.124	<0.0001
**O2**	0.938	0.930	−0.007	0.825
**P4**	0.925	0.935	0.01	0.088
*** P8**	0.887	0.933	0.046	<0.0001
**TP10**	0.936	0.933	−0.003	0.625
**CP6**	0.930	0.936	0.005	0.283
**CP2**	0.923	0.933	0.009	0.253
**Cz**	0.936	0.930	−0.005	0.687
**C4**	0.932	0.935	0.002	0.452
**T8**	0.930	0.935	0.004	0.35
**FT10**	0.938	0.928	−0.01	0.832
**FC6**	0.934	0.929	−0.004	0.691
**FC2**	0.936	0.935	0.0	0.524
**F4**	0.934	0.930	−0.003	0.645
**F8**	0.932	0.930	−0.001	0.679
**Fp2**	0.933	0.932	0.0	0.58

## Data Availability

Data available upon request.

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
