# Peer review of "The Role of Stimuli-Driven and Goal-Driven Attention in Shopping Decision-Making Behaviors—An EEG and VR Study"

_brainsci, 2023, doi:10.3390/brainsci13060928_

Round 1

Reviewer 1 Report

This paper examines goal-directed and stimulus-driven attention using EEG measures during a Virtual Reality task. The study presents new evidence on the modulation of brain activity in an ecologically valid setting under different attentional modes (goal or stimulus-driven attention). The results were obtained using relatively new analytical EEG methodologies in a VR context. During planned purchases compared to unplanned purchases and rest, increased activity was observed in the alpha and theta bands over frontal and parietal lobes, while the opposite pattern was observed during unplanned purchases. Frontoparietal connectivity increased during planned purchases, while SpE increased in these regions during the unplanned condition. These results have implications for understanding consumer behavior and may provide a reference for future research on how to analyze and interpret EEG signals recorded during VR for a more ecological evaluation of attention that can be applied to various fields.

1.- Having said that, my Main Comment concerns the interpretation of the cognitive process and brain activity observed in the Unplanned Purchase condition. While it is reasonable to associate the Planned Purchase condition with the assumed goal-directed mechanisms, as detailed in the Introduction, it is much more challenging to determine the type of attentional activity occurring during Unplanned Purchase in the absence of any behavioral information about the participants' "free decisions." Were individuals asked about their decisions in this condition? Could "perceptual salience" or any other measured factor a significant predictor of individuals' performance, as suggested in the Introduction? Note that the bottom-up/top-down nature of the attentional processes involved may vary depending on such factors. Without additional evidence, the requirement to direct attention towards external stimuli during Unplanned Purchase does not necessarily warrant the classification of the attentional mode as stimulus-driven or bottom-up. I assume that individuals' choices were recorded at some point during task performance. Therefore, providing additional information on the type of decisions made may lend further support to the authors' attentional assumptions regarding the Unplanned Purchase

Here are some additional Minor Comments:

(Introduction)

2. On page 3, line 115, the correct term may be bottom-up (not bottom-down).

3.- On page 4, lines 172-173, the hypothesis regarding SpE could be better described. Specifically, it is not clear what “an alternation of SpE in frontal, parietal, and occipital lobes between goal-directed and stimulus-driven attention” means at a data analysis level.

(Method)

4. Please, provide a more operational definition or description of the "bad" ICA components that were removed from the analyses on page 5, line 230. For example, it could be useful to provide details about how many components were eliminated on average for each subject or how the distribution of electrical fields was in these cases.

Author Response

Dear Reviewer

I have submitted the revised version of the manuscript and response letter for your comments.

Sincerely yours.

Reviewer 2 Report

The article presents an EEG study conducted on 29 participants regarding the dynamics between bottom-up and top-down processes during purchasing processes. During the study, participants immersed in a simulated virtual reality supermarket had to purchase products from a predefined list and then choose products of their choice. The researchers thus realized two conditions to which each participant was subject: a planned purchase condition and an unplanned purchase condition. They then hypothesized that the planned condition activated more bottom-up attentional processes and, therefore, the related EEG correlates. The article is interesting and informative; however, some points deserve attention:

- The introduction, particularly the first paragraph, is rather simple and introduces the work as if it were a chapter in a psychology textbook, rather than a scientific article

- The authors say that the goal of the work is to analyze the EEG of people in real-life situations. In reality, the proposed tasks are within a virtual environment within which the subjects move artificially and in a rather rigid setting. Therefore, this is not a real-life study.

- The authors analyzed the VR experience qualitatively in order to find any correlations between, for example, sense of presence or discomfort and outcomes?

- The authors report values of different times devoted to the planned and unplanned condition, but they do not specify how much money was allocated between the two conditions. Since the unplanned condition can be thought of as being freer and more time-consuming, I imagine that there was little money left and therefore few choices made.

- The authors correlate their results to bottom-up and top-down attentional processes, but how do they separate the effects of the different mental and behavioral operations required to perform such different tasks? In practice, what do the detected EEG differences relate to?

- The authors report in section 3.4 "classification results" something that was not described either in the objectives or in the statistical analysis section. What is the purpose of this further analysis?

- Given the points above, the discussion of the results appears speculative at times and does not discuss further possible interpretations.

- In general, the paper needs significant editing.

Author Response

(The authors gave the same response as above.)

Reviewer 3 Report

The authors aim to assess three EEG-based measures of attention (Power Spectral Density, Connectivity, and Spectral Entropy) in decision-making situations involving goal-directed and stimulus-driven attention using a Virtual Reality supermarket. The idea is very interesting and the paper is well writen, with some minors errors needing attention and listed in sequence. Please note that the following comments are intended to improve paper quality and readers' understanding.

The study compares two moments: a moment regarding "planned purchase" and another regarding "unplanned purchase". Philosophicaly speaking, what do the authors think if the same experiment was performed on a real supermarket? Would the captured values be different? Why? I believe some considerations about that should be in the text.

I know that the authors performed a high-level analysis, but did you perceived any difference according on the captured signals according to the product being bought? If this is relevant, please make some considerations about that as well.

I believe the conclusion section is too small for such a rich paper. Please increase the size of the conclusion section providing more details about the main contributions of the paper and how it advances the state of the art. Also provide more future work.

More general comments and minors errors are listed as follows.

"Abstract :The" -> "Abstract: The"

"is, similar" -> "is similar"

"features and properties of shopping items and their properties" -> please rewrite

"shelf placement on shelves," -> please rewrite

why is the entire "Theta activity" topic in italic?

"2021)and" -> "2021) and"

"behaviors, have" -> "behaviors have"

" stimuli, has" -> " stimuli has"

"system. of these" -> "system. Of these"

"them in the VR" -> "them in VR"

"they want" -> "they wanted"

"money, was" -> "money was"

" expected that" -> " expected"

"alloctaed" -> "allocated"

"Functional Connectivity." -> "Functional Connectivity"

"conduction(Friston" -> "conduction (Friston"

"the final connectivity" -> ", the final connectivity"

"welch method" -> "Welch's method"

"networks(Corbetta " -> "networks (Corbetta "

"2002c),  our" -> "2002c), our"

Some figures are too big. Please resize them.

"parietal-parietal) the" -> "parietal-parietal), the"

"Entripy2 " -> "Entropy 2 "

Table 3 is too big. Please decrease its font size.

"with ×. ." -> "with ×."

"was 0.79 ± 0.05was 0.79 ± 0.05." -> "was 0.79 ± 0.05."

" tasks(Ratcliffe" -> " tasks (Ratcliffe"

" in Frontal, and parietal Lobes" -> " in Frontal and parietal Lobes"

"Whereas." -> ?

"Increases synchronized activity over the frontoparietal network in theta for goal-driven attention." -> "Increased synchronized activity over the frontoparietal network in theta for goal-driven attention"

"Kastner et al., 1999) " -> "Kastner et al., 1999)."

"the role SpE" -> "the SpE role"

" model, would" -> " model would"

"combined.  In" -> "combined. In"

" more generalize" -> " more generalized"

" limitations worth" -> " limitations are worth"

Author Response

(The authors gave the same response as above.)

Round 2

Reviewer 2 Report

The authors did an excellent job in revising their paper. Consequently, I feel that the article can now be published.